# A-Kinase Anchoring Proteins Diminish TGF-β_1_/Cigarette Smoke-Induced Epithelial-To-Mesenchymal Transition

**DOI:** 10.3390/cells9020356

**Published:** 2020-02-03

**Authors:** Haoxiao Zuo, Marina Trombetta-Lima, Irene H. Heijink, Christina H. T. J. van der Veen, Laura Hesse, Klaas Nico Faber, Wilfred J. Poppinga, Harm Maarsingh, Viacheslav O. Nikolaev, Martina Schmidt

**Affiliations:** 1Department of Molecular Pharmacology, University of Groningen, 9713AV Groningen, The Netherlands; haoxiaozuo1212@gmail.com (H.Z.); m.trombetta.lima@rug.nl (M.T.-L.); c.h.t.j.van.der.veen@rug.nl (C.H.T.J.v.d.V.); wilfredpoppinga@gmail.com (W.J.P.); 2Groningen Research Institute for Asthma and COPD, GRIAC, University Medical Center Groningen, University of Groningen, 9713AV Groningen, The Netherlands; h.i.heijink01@umcg.nl (I.H.H.); l.hesse@umcg.nl (L.H.); 3Institute of Experimental Cardiovascular Research, University Medical Centre Hamburg-Eppendorf, 20246 Hamburg, Germany; v.nikolaev@uke.de; 4Department of Pathology and Medical Biology Groningen, University Medical Center Groningen, University of Groningen, 9713AV Groningen, The Netherlands; 5Department of Pulmonology, University Medical Center Groningen, University of Groningen, 9713AV Groningen, The Netherlands; 6Department of Gastroenterology and Hepatology, University Medical Center Groningen, University of Groningen, 9713AV Groningen, The Netherlands; k.n.faber@umcg.nl; 7Department of Pharmaceutical Sciences, Lloyd L. Gregory School of Pharmacy, Palm Beach Atlantic University, West Palm Beach, FL 33401, USA; HARM_MAARSINGH@pba.edu; 8German Center for Cardiovascular Research (DZHK), 20246 Hamburg, Germany

**Keywords:** epithelial-to-mesenchymal transition, TGF-β1, cAMP, A-kinase anchoring protein, Ezrin, AKAP95, Yotiao, cigarette smoke, COPD

## Abstract

Epithelial-to-mesenchymal transition (EMT) plays a role in chronic obstructive pulmonary diseases (COPD). Cyclic adenosine monophosphate (cAMP) can inhibit transforming growth factor-β1 (TGF-β1) mediated EMT. Although compartmentalization via A-kinase anchoring proteins (AKAPs) is central to cAMP signaling, functional studies regarding their therapeutic value in the lung EMT process are lacking. The human bronchial epithelial cell line (BEAS-2B) and primary human airway epithelial (pHAE) cells were exposed to TGF-β1. Epithelial (E-cadherin, ZO-1) and mesenchymal markers (collagen Ӏ, α-SMA, fibronectin) were analyzed (mRNA, protein). ELISA measured TGF-β1 release. TGF-β1-sensitive AKAPs Ezrin, AKAP95 and Yotiao were silenced while using siRNA. Cell migration was analyzed by wound healing assay, xCELLigence, Incucyte. Prior to TGF-β1, dibutyryl-cAMP (dbcAMP), fenoterol, rolipram, cilostamide, and forskolin were used to elevate intracellular cAMP. TGF-β1 induced morphological changes, decreased E-cadherin, but increased collagen Ӏ and cell migration, a process that was reversed by the inhibitor of δ/epsilon casein kinase I, PF-670462. TGF-β1 altered (mRNA, protein) expression of Ezrin, AKAP95, and Yotiao. St-Ht31, the AKAP antagonist, decreased E-cadherin (mRNA, protein), but counteracted TGF-β1-induced collagen Ӏ upregulation. Cigarette smoke (CS) increased TGF-β1 release, activated TGF signaling, augmented cell migration, and reduced E-cadherin expression, a process that was blocked by TGF-β1 neutralizing antibody. The silencing of Ezrin, AKAP95, and Yotiao diminished TGF-β1-induced collagen Ӏ expression, as well as TGF-β1-induced cell migration. Fenoterol, rolipram, and cilostamide, in AKAP silenced cells, pointed to distinct cAMP compartments. We conclude that Ezrin, AKAP95, and Yotiao promote TGF-β1-mediated EMT, linked to a TGF-β1 release by CS. AKAP members might define the ability of fenoterol, rolipram, and cilostamide to modulate the EMT process, and they might represent potential relevant targets in the treatment of COPD.

## 1. Introduction

Chronic obstructive pulmonary disease (COPD), which is primarily induced by cigarette smoke (CS), is characterized by irreversible airflow limitation that is linked to subepithelial airway fibrosis [1]. A vital player during organ fibrosis is epithelial-to-mesenchymal transition (EMT), a process in which epithelial cells gradually lose their epithelial phenotype and undergo transition to typical mesenchymal characteristics, which feature increased mitogenic capacity and enhanced extracellular matrix production [2,3,4,5]. Recent evidence suggests that EMT is involved in the fibrotic processes in the large and small airways during the pathogenesis of COPD as well as lung cancer [6,7,8]. Importantly, studies have provided evidence that EMT is an active process in the airways of smokers, particularly in those current-smoking COPD patients, which indicates that CS-induced EMT contributes to COPD pathogenesis [6,7].

Transforming growth factor-β1 (TGF-β1) is another well-known inducer of EMT [5,9]. The cyclic adenosine monophosphate (cAMP) signaling pathway is one of the multiple pathways that are implicated in the regulation of EMT [10,11,12,13]. A-kinase anchoring proteins (AKAPs) are a group of structurally diverse proteins that are localized at specific subcellular sites. They play a critical role in maintaining subcellular compartmentalization of cAMP by generating spatially discrete signaling complexes, which create local gradients of cAMP [14,15]. As scaffolding proteins, AKAPs bind protein kinase A (PKA) and a diverse subset of signaling enzymes, and thereby control cAMP microdomains in a spatio-temporal manner [5,16]. Existing studies have demonstrated that several AKAPs members are involved in TGF-β1-induced EMT in vitro. For instance, suppressing the expression of the AKAP family member Ezrin by small interfering RNA reduced both morphological changes and TGF-β1-induced EMT cell migration in human alveolar A549 cells [17]. Knockdown by short hairpin RNA of another AKAP member Yotiao, which is also known as AKAP9, inhibited TGF-β1-induced EMT in colorectal cancer cells [18]. Additionally, AKAP9 interacts and co-localizes with E-cadherin at the cell membrane [19]. More importantly, the silencing of AKAP9 reduced the functional epithelial barrier, which suggests the possibility of a specific role for AKAP9 in the maintenance of the epithelial barrier [19]. However, the function of AKAPs in regulating TGF-β1/CS-induced EMT in human bronchial epithelial cells is still unclear.

In the present study, we hypothesized that AKAPs could regulate TGF-β1/CS-induced EMT in human bronchial epithelial BEAS-2B cells. We found that collagen I upregulation that is induced by TGF-β1 is diminished when AKAP-PKA interactions were disrupted by st-Ht31, whereas TGF-β1-induced E-cadherin downregulation was not reversed by st-Ht31, which indicated that AKAPs are selectively involved in TGF-β1-induced collagen I increase. CS exposure increased TGF-β1 release and activated the TGF-β1 signaling pathway, which could be blocked by TGF-β1 neutralizing antibodies, therefore contributing to EMT progression. We observed that mRNA and protein expression of the three AKAPs members Ezrin, Yotiao, and AKAP95 was changed after TGF-β1 stimulation. The co-silencing of Ezrin, AKAP95, and Yotiao inhibited TGF-β1-induced cell migration in BEAS-2B cells and primary human airway epithelial cells. In addition, co-silencing of Ezrin, AKAP95, and Yotiao specifically accelerated the β_2_-adrenergic receptor (β_2_-AR)-induced reduction in TGF-β1-induced collagen Ӏ upregulation. The effects of cilostamide and rolipram were largely left unchanged pointing to AKAP defined cAMP sub-compartments.

## 2. Materials and Methods

### 2.1. Chemicals and Antibodies

Recombinant human TGF-β1 protein was from R&D systems (Abingdon, UK). Fenoterol was purchased from Boehringer Ingelheim (Ingelheim, Germany). Rolipram, cilostamide, and bovine serum albumin (BSA) were from Sigma-Aldrich (St-Louis, MO, USA). Forskolin was from Tocris Bioscience (Bristol, UK). InCELLect™ AKAP St-Ht31 inhibitor peptide was purchased from Promega (Leiden, the Netherlands). The transfect reagent lipofectamine RNAiMax was purchased from Invitrogen (Bleiswijk, the Netherlands). Control siRNA, Ezrin siRNA, AKAP95 siRNA, and Yotiao siRNA were obtained from Santa Cruz Biotechnology (Heidelberg, Germany). All other chemicals were of analytical grade. Table 1 lists the origin and dilution of the antibodies used.

### 2.2. Cell Culture

The human bronchial epithelial cell line BEAS-2B was maintained in RPMI 1640 that was supplemented with 10% *v*/*v* heat-inactivated fetal bovine serum (FBS) and antibiotics (penicillin 100 U/mL, streptomycin 100 μg/mL) in a humidified atmosphere of 5% (*v*/*v*) CO_2_ at 37 °C. The cells were dissociated from T75 flasks with trypsin/EDTA and seeded in appropriate cell culture plates. Cells were maintained in 1% *v*/*v* FBS medium 24 h before and during stimulation, since a serum-free medium induced cell death.

Primary human airway epithelial (pHAE) cells were isolated from residual tracheal and main stem bronchial tissue, from lung transplant donors post-mortem, within 1–8 h after lung transplantation, while using the selection criteria for transplant donors according to the Eurotransplant guidelines. The tracheal tissue was collected in a Krebs-Henseleit buffer (composition in mM: NaCl 117.5, KCl 5.6, MgSO_4_ 1.18, CaCl_2_ 2.5, NaH_2_PO_4_ 1.28, NaHCO_3_ 25, and glucose 5.5) and primary HAE cells were collected by enzymatic digestion, as previously described [20]. In short, the airway epithelial cells were gently scraped off the luminal surface, washed once, and then submerged cultured on petri-dishes that were pre-coated with a combination of fibronectin (10 μg/mL), bovine type I collagen (30 µg/mL), and bovine serum albumin (10 μg/mL) in PBS, while using a keratinocyte serum free medium (Gibco, Carlsbad, CA, USA) that was supplemented with 25 µg/mL bovine pituitary extract, 0.2 ng/mL epidermal growth factor, and 1 μM isoproterenol for 4–7 days until they reached confluence, and were then trypsinized and seeded into six-well plates for silencing experiments.

### 2.3. Cell Stimulation

The BEAS-2B cells were grown to confluence and then starved by exchange of complete medium to 1% *v*/*v* FBS medium for 24 h. Cells were treated with 1 ng/mL, 3 ng/mL, and 10 ng/mL for 24 h, 48 h, and 72 h. 3 ng/mL TGF-β1 treatment for 24 h was used in current study based on gene and protein expression of EMT markers. Cells were pretreated for 30 min. before stimulation with TGF-β1 for 24 h with st-Ht31 (50 μM) to disrupt AKAP-PKA interaction [21] or with the casein kinase 1δ/ε inhibitor PF-670462 (1 and 10 μM) [22] to confirm that TGF-β1-induced EMT could be reversed in BEAS-2B cells. The β_2_-agonist fenoterol (0.001–10 μM), the phosphodiesterase (PDE4) inhibitor rolipram (1 or 10 μM), the PDE3 inhibitor cilostamide (10 μM), and adenylyl cyclase agonist forskolin (10 μM) were added 30 min. without TGF-β1, followed by 24 h stimulation with TGF-β1. Different concentrations of CS extract were used to stimulate cells for 24 h and supernatant was collected for measuring TGF-β1 production by ELISA and incubating basal BEAS-2B cells for 1 h. The concentrations of TGFβ1 in cell supernatants were determined while using ELISA according to manufacturer’s protocol (DY240 and DY010, R&D Systems, BioTechne, Minneapolis, MN, USA).

### 2.4. Transfection

The cells were grown to 80% confluence and were then transfected while using lipofectamine RNAiMax reagent in a 1:1 reagent: siRNA ratio in complete growth medium without antibiotics. Cells were transfected with 40 nM control siRNA, 40 nM Ezrin siRNA, 40 nM AKAP95 siRNA, and 40 nM Yotiao siRNA for 48 h before TGF-β1 treatment. After TGF-β1 treatment for 24 h, the cells were lysed for real-time quantitative PCR and western blotting analysis.

### 2.5. Real-Time Quantitative PCR

Total RNA was extracted from cells while using the Maxwell 16 LEV simplyRNA Tissue Kit (Promega, Madison, WI, USA), according to the manufacturer’s instructions. The total RNA yield was determined by NanoDrop 1000 Spectrophotometer (Thermo Fisher Scientific, Wilmington, DE, USA). Equal amounts of RNA were used to synthesize cDNA. An Illumina Eco Real-Time PCR system was used to perform real-time qPCR experiments. PCR cycling was performed with denaturation at 94 °C for 30 s, annealing at 59 °C for 30 s, and extension at 72 °C for 30 s for 45 cycles. RT-qPCR data was analyzed by LinRegPCR software [23]. The amount of target gene was normalized to the reference genes 18S ribosomal RNA, SDHA, and RPL13A to analyze RT-qPCR data. Table 2 lists primer sequences.

### 2.6. Western Blotting

Cells were lyzed in a lysis buffer and BCA protein assay (Pierce) measured homogenized protein concentration. Equal amounts of the total proteins were loaded into 10% SDS-polyacrylamide gel electrophoresis. After being transferred to a nitrocellulose membrane, the membranes were blocked with Roti-Block (Carl Roth, Karlsruhe, Germany). Primary antibodies (Table 1) were incubated at 4 °C overnight, followed by a secondary antibody (anti-mouse IgG, 1:5000, anti-rabbit IgG, 1:5000, or anti-goat IgG, 1:5000, Sigma) incubation at room temperature for two hours. The protein bands were developed on film while using a Western detection ECL-plus kit (PerkinElmer, Waltman, MA, USA). ImageJ software was used for band densitometry analysis.

### 2.7. Immunofluorescence

50,000 cells were seeded on coverslips (12 mm) and then stimulated with different reagents for a certain period, as described previously. Subsequently, the cells were fixed with 1:1 methanol/acetone in at −20 °C for 20 min. After 3 times washing with PBS, the cells were then blocked using 1% (*w*/*v*) BSA/PBS with 2% donkey serum for one hour. Primary antibodies (−) were applied overnight at 4 °C, after which secondary antibody Alexa Fluor 488 nm donkey anti-goat and Cy™3 AffiniPure donkey anti-mouse (Jackson, Cambridgeshire, UK) were incubated for 2 h. Finally, the slides were mounted with a mounting medium containing DAPI (Abcam, Cambridge, UK). The images were captured with a Leica DM4000b Fluorescence microscope (Leica Microsystems, Germany) that was equipped with a Leica DFC 345 FX camera.

### 2.8. Cigarette Smoke Extract (CSE) Preparation

CSE preparation was prepared as previously described [21]. Two 3R4F research cigarettes (University of Kentucky, Lexington, KY, USA) without a filter were combusted into 25 mL 1% *v*/*v* FBS medium while using a peristaltic pump (45 rpm, Watson Marlow 323E/D, Rotterdam, The Netherlands). This medium was designated as 100% CSE and it was diluted to certain concentrations in different experiments.

### 2.9. Wound Healing Assay

The cells were grown to confluence and scratched with a pipette tip. After being washed once to remove the detached cells, cells were allowed to migrate into the wound area in the absence or presence of TGF-β1 and different siRNAs. The wound area was photographed immediately after a scratch and then after 24 h of stimulation.

### 2.10. xCELLigence Transwell Migration

Cell migration was further tested while using the label-free and real-time xCELLigence transwell migration system CIM-16 plates (RTCA DP, ACEA Biosciences, San Diego, CA, USA). A 10% *v*/*v* FBS growth medium was applied as a chemoattractant in the lower chamber. 25 μL of 1% *v*/*v* FBS medium was added to the top chamber and the plates were placed in the system for equilibration. Cells were passaged and 90,000 cells were seeded on the top chamber in 1% *v*/*v* FBS medium containing TGF-β1 or CS extract. Cells were then placed back in the system for future monitoring for 24 h at 37 °C in a 5% (*v*/*v*) CO_2_ humidified atmosphere. The system was set to take a cell index measurement at 15 min. intervals.

### 2.11. Incucyte

The BEAS-2B cells were transduced with NucLight Red lentivirus (Essen Bioscience, Ann Arbor, MI, USA) to produce red fluorescent proteins that label the BEAS-2B cell nucleus according to the product instruction. Red-labeled BEAS-2B cells (10,000 per well) were plated on 96-well ImageLock cell migration plates (Essen Bioscience) and then incubated overnight. After silencing with a combination of Ezrin, AKAP95 and Yotiao siRNA, the cell monolayer was scratched with a 96-pin WoundMaker (Essen Bioscience), and the cells washed with PBS (phosphate-buffered saline) before adding 3 ng/mL TGF-β1 or diluted CS extract. Cell migration and proliferation were monitored by a microscope gantry inside a cell incubator, which was connected to a networker external controller hard drive that gathered and processed image data (Incucyte, Essen Bioscience).

### 2.12. Cell Viability

In BEAS-2B and pHAE cells, cell viability was determined based on metabolic activity while using the 3-(4,5-dimethylthiazol-2-yl)-2,5-diphenyltetrazolium bromide assay (MTT; Sigma Aldrich, Zwijndrecht, the Netherlands) at a final concentration of 0.5 g/L, by incubation for 1 h at 37 °C and 5% CO_2_. Subsequently, MTT solution was removed, the cells were kept for at least 1 h at −20 °C, and DMSO was used to solubilise the formed formazan. The absorbance of each well was determined with the Synergy H1 Multi-Mode reader (Biotek, Louisanacity, LA, USA) at 570 nm and at 630 nm for background subtraction. The values of non-treated control cells were normalized to 100% with which values of the treated cells were compared. Data were collected from six wells per condition.

### 2.13. Cell Cycle Distribution

BEAS-2B and pHAE cells were grown to 80% confluence and they were transfected using lipofectamine RNAiMax in a 1:1 reagent: siRNA ration in complete growth medium without antibiotics. As described under 2.4, the cells were transfected with 40 nM control siRNA or with a mix of 40 nM Ezrin siRNA, 40 nM AKAP95 siRNA, and 40 nM Yotiao siRNA for 48 h. After transfection, cells were treated with 3 ng/mL TGF-β1 and/or 1% CSE for 24 h. After treatment, cells were trypsinized and fixated with a 95% ethanol PBS solution for 72 h. Total DNA from the cells was stained for 15 min. using a 100 µg/mL propidium iodide PBS solution. After staining, cells were lysed in PBS. Fluorescent excitation of propidium iodide was determined by flow cytometry while using the CytoFLEX S benchtop flow cytometer for at least 10,000 cells (Beckman Coulter Life Sciences, Indianapolis, IN, USA). The quantification of the data was acquired using FlowJoTM (BD, Advancing the World of Health, Ashland, OR, USA).

### 2.14. Statistics

All of the data were analyzed by GraphPad Prism (GraphPad Software, Inc., San Diego, CA, USA) and expressed as mean ± SEM. At least three independent experiments were conducted for each treatment. The exact repeats are indicated in the figure legends. The Shapiro-Wilk test determined normal data distribution. The statistical significance was performed while using unpaired Student’s t-test or ANOVA test for multiple comparisons. For non-Gaussian distributed data, the significance was determined using a non-parametric one-way ANOVA with a post hoc Kruskal-Wallis multiple comparisons test. A *p* < 0.05 was considered to be statistically significant.

## 3. Results

### 3.1. Effect of TGF-β1 on Cell Morphology and Phenotype Markers in BEAS-2B Cells

As shown in Figure 1A, TGF-β1 stimulation for 24 h changed the morphology of BEAS-2B cells from a typical epithelial shape to an elongated and spindle-like morphology. TGF-β1 induced an increase in phospho-PKA substrates that was indicative of a global elevation in cellular cAMP (Appendix A). The mRNA expression of *E-cadherin* was significantly decreased in TGF-β1 stimulated cells as compared to the control cells, whereas TGF-β1 dramatically up-regulated *collagen Ӏ* mRNA expression (Figure 1B). The protein levels of multiple epithelial and mesenchymal markers were analyzed by western blotting, including collagen Ӏ, fibronectin, and α-smooth muscle actin (α-SMA) (Figure 1C). Following the effect of mRNA levels, TGF-β1 strongly decreased E-cadherin, while increasing collagen I protein levels (Figure 1C). The signals of fibronectin and α-SMA were weaker and more variable as compared to collagen Ӏ (Figure 1D). Thus, collagen Ӏ and E-cadherin were used as markers for the mesenchymal and epithelial phenotype makers from this point, respectively. In addition, immunofluorescence images showed that the protein expression of another epithelial marker ZO-1 was significantly decreased after TGF-β1 stimulation, whereas collagen Ӏ protein expression was clearly enhanced (Figure 1D). The immunofluorescent staining of E-cadherin was not as obvious as that of ZO-1, however a clear decrease in the protein expression of E-cadherin was observed after TGF-β1 stimulation (Figure 1E).

### 3.2. Disruption of AKAP-PKA Interaction Diminishes TGF-β1-Induced Collagen Ӏ Upregulation

The effect of the cell-permeable PKA-anchoring disruptor peptide, st-Ht31, on gene and protein expression of EMT markers was examined to determine the role of the physical interaction between AKAP and PKA in the TGF-β1-induced EMT. As shown in Figure 2A, treatment with 50 μM st-Ht31 alone significantly decreased *E-cadherin* gene expression and this effect was even more pronounced at the protein level (Figure 2C).

Consequently, st-Ht31 pre-treatment did not prevent the TGF-β1-induced downregulation of E-cadherin (Figure 2A,C). In contrast, st-Ht31 significantly decreased collagen Ӏ protein expression in TGF-β1-stimulated cells (Figure 2D), leaving mRNA levels unchanged (Figure 2B). TGF-β1 induced the EMT-driving transcription factors Snail and Slug (Figure 2E).

### 3.3. CSE Activates TGF-β1 Signaling Pathway

It has been demonstrated that cigarette smoke (CS) exposure induces TGF-β1 release [6,24] and may therefore contribute to EMT in lung epithelial cells. Indeed, we confirmed that CSE exposure significantly increases TGF-β1 release by BEAS-2B cells that were used in this study (Figure 3A). Moreover, the phosphorylation of SMAD2 (Figure 3B) and SMAD3 (data not shown) was increased by CSE exposure. We next used TGF-β neutralizing antibodies to block TGF-β1 signaling to confirm that CSE-induced EMT depends on TGF-β1 release. A 24 h exposure of BEAS-2B cells to 1% CSE significantly decreased E-cadherin protein levels, which was reversed when TGF-β neutralizing antibodies were added 30 min. prior the TGF-β challenge (Figure 3C). To test whether the TGF-β1-induced E-cadherin decrease could be modulated in the BEAS-2B cells, these cells were exposed to a selective inhibitor of the δ- and ε-isoforms of casein kinase I, PF-670462, as it was previously shown to reverse TGF-β1-induced EMT in A549 cells [22]. Indeed, the pretreatment of BEAS-2B cells with PF-670462 dose-dependently prevented TGF-β1-induced E-cadherin loss and collagen Ӏ gain (Figure 3E), indicating that δ- and ε-isoforms of casein kinase I and AKAPs distinctly regulate the TGF-β1-induced EMT in BEAS-2B cells.

### 3.4. Ezrin, AKAP95, and Yotiao Are Involved in TGF-β1-Induced EMT

Afterwards, we studied which member(s) of the AKAP family exert sensitivity to TGF-β1 in BEAS-2B cells. TGF-β1 selectively and significantly down-regulated the mRNA levels of *Ezrin*, whereas the mRNA expression of *AKAP95* and *Yotiao* was enhanced. *AKAP1*, *AKAP5*, *AKAP11*, and *AKAP12* mRNA levels were not affected by TGF-β1 (Figure 4A). We observed an increase in Yotiao and AKAP95 protein, but not Ezrin (Figure 4B). Immunofluorescence microscopy staining also showed that TGF-β1 significantly increased the protein expression of Yotiao (Figure 4C).

As TGF-β1 modulated the expression of Ezrin, AKAP95, and Yotiao, we hypothesized that these factors might be involved in TGF-β1-induced EMT. To study this, we silenced the expression of Ezrin, AKAP95, and Yotiao in BEAS-2B cells while using small interfering RNAs (siRNA). Real-time quantitative PCR confirmed the siRNA-mediated reduction of *Ezrin*, *AKAP95*, and *Yotiao* (29.4 ± 7.4%, 34.4 ± 6.8%, and 43.4 ± 15.3%, respectively;), which was accompanied with similar reductions only in the specific corresponding proteins (Ezrin, 25.1 ± 0.1%; AKAP95 51.2 ± 3.7%; Appendix A). We observed 40% and 63% decreased E-cadherin protein levels in Ezrin- or AKAP95-silenced cells, respectively, an effect that was even more pronounced in TGF-β1-treated cells (Figure 5A, Table 3). The silencing of Yotiao did not reduce the E-cadherin expression (Figure 5A). Silencing of Ezrin, AKAP95, or Yotiao suppressed TGF-β1-induced upregulation of the mesenchymal maker collagen Ӏ by about 40% (Figure 5B, Table 3). We then questioned whether the co-silencing of Ezrin, AKAP95, and Yotiao would further alter the expression of the EMT markers. The co-silencing of all three factors (siM, multiple silencing) reduced the protein levels of E-cadherin after TGF-β1 stimulation to 11% (Figure 5C, Table 3), as compared to ~40% when the factors were individually silenced (Figure 5A). More importantly, the collagen Ӏ protein levels were reduced to 25% after triple-silencing in TGF-β1-stimulated cells when compared to ~60% after single-silencing (Figure 5D, as compared to Figure 5B, Table 3). Therefore, we performed the next experiments in cells with co-silenced Ezrin, AKAP95, and Yotiao (referred to siM).

### 3.5. Ezrin, AKAP95, and Yotiao Are Required for TGF-β1-Induced Cell Migration

As expected, TGF-β1 stimulation increased BEAS-2B cell motility when compared to control cells, as analyzed in scratch assays (Figure 6A–D). Treatment with 50 μM st-Ht31 profoundly reduced cell migration and normalized cell migration of TGF-β1-stimulated cells back to control levels (Figure 6A,B). Similarly, the co-silencing of Ezrin, AKAP95, and Yotiao profoundly reduced cell migration and normalized cell migration of TGF-β1-stimulated cells back to the control levels (Figure 6C,D). In contrast, the co-silencing of Ezrin, AKAP95, and Yotiao did neither profoundly alter cell viability nor the cell cycle distribution of cells that were stimulated with TGF-β1 or CSE (Figure 6E,F).

The real-time system Incucyte monitored cell migration. The migration of cells with co-silenced Ezrin, AKAP95, and Yotiao upon wounding was significantly slowed down both at baseline and upon treatment with TGF (Figure 7A). In silenced cells, TGF-β1 increased cell migration was significantly slower when compared to cells. Additionally, we found that the cell proliferation within 24 h in each treatment was quite limited, which indicated that the wound closure was due to migration instead of proliferation (Figure 7B). The effect of CS extract exposure on activating cell migration was shown by the real time monitoring system Incucyte (Figure 7C,D). In another real time assay for cell migration using the xCELLigence platform, TGF-β1 increased cells migration in the early phase, which was reduced in cells that were co-treated with the siRNA of Ezrin, AKAP95 and Yotiao (Appendix A). Additionally, we found that CSE exposure enhanced cell migration in the early phase, which was examined by the xCELLigence transwell system. The co-silencing of Ezrin, AKAP95, and Yotiao decreased CS-induced cell migration (Appendix A).

### 3.6. The Role of AKAPs in Primary HAE Cells

We applied identical treatments to primary human airway epithelial (HAE) cells to translate our findings obtained using the BEAS-2B cells to clinically more relevant cell types. As shown in Figure 8A, exposure to CSE increased TGF-β1 mRNA. TGF-β1 specifically and significantly increased the expression of *Yotiao*, *Ezrin* (Figure 8B). TGF-β1 strongly decreased the protein expression of E-cadherin, but increased the protein expression of collagen I (Figure 8C,D). The co-silencing of Ezrin, AKAP95, and Yotiao diminished the TGF-β1-induced upregulation of collagen I, leaving the protein expression of E-cadherin unchanged (Figure 8C,D). In addition, we found that TGF-β1 was able to decrease cell-cell interaction, which was observed in immunofluorescence staining of ZO-1 (Figure 8D).

Additionally, we also investigated whether silencing three AKAP genes could affect the cell migration using primary HAE cells (pHAE) to confirm the findings in BEAS-2B cells. Similar results were observed in pHAE cells, even though the overall migration in primary HAE cells was much less when compared with that in BEAS-2B cells, as shown in Figure 9A,B. Importantly, TGF-β1 was no longer able to promote cell migration in cells silenced Ezrin, AKAP95, and Yotiao. Of note was that silencing of the TGF-β1 sensitive AKAPs did not interfere with the basal migration capacity in primary epithelial cells. The co-silencing of Ezrin, AKAP95, and Yotiao did not profoundly alter cell viability and the cell cycle distribution of cells stimulated with TGF-β1 (Figure 9C,D).

### 3.7. cAMP Donors Decrease TGF-β1-Induced Collagen Ӏ Upregulation

The cell-membrane permeable cAMP derivative dbcAMP was used to disrupt cAMP compartmentalization in BEAS-2B cells to further study the role of compartmentalized cAMP. We found that dbcAMP dose-dependently increased the protein expression of epithelial marker E-cadherin in control BEAS-2B cells, without affecting the reduced levels of E-cadherin in the TGF-β1-treated cells (Figure 10A). In contrast, TGF-β1-induced collagen Ӏ upregulation was significantly decreased by dbcAMP in a dose dependent manner, while leaving the basal levels unaffected (Figure 10B). Immunofluorescence microscopy analyses revealed that dbcAMP slightly reduced the basal expression of ZO-1 in BEAS-2B cells, while leaving the TGF-β1-induced reduction largely unaffected (Figure 10C,D). Moreover, TGF-β1-induced collagen I expression was reduced by dbcAMP in these cells, without affecting the basal levels (Figure 10C,D).

The effect of the β_2_-agonist fenoterol, the PDE4 inhibitor rolipram, the PDE3 inhibitor cilostamide, and the adenylyl cyclase activator forskolin were studied to further evaluate whether cAMP compartmentalization contributed to the EMT process in our model. Although none of these compounds affected basal levels of E-cadherin or the reduction induced by TGF-β1 (Figure 11A,C), they each suppressed TGF-β1-induced collagen Ӏ upregulation (Figure 11B,D).

### 3.8. Ezrin, AKAP95, and Yotiao Differentially Contribute to cAMP Compartments

We tested the effects of fenoterol, rolipram, and cilostamide in Ezrin-AKAP95-Yotiao (siM) co-silenced cells in order to study to which extent defined cAMP compartments might contribute to the TGF-β1-induced EMT process in BEAS-2B cells. We found that fenoterol further decreased siM-induced collagen Ӏ downregulation from 54.5 ± 9.1% to 24.9 ± 8.0% (Figure 12A, Table 4). On the contrary, rolipram and cilostamide were unable to further reduce collagen Ӏ protein expression (Figure 12B,C, Table 4), indicating that Ezrin, AKAP95, and Yotiao were associated with β_2_-AR in decreasing TGF-β1-induced collagen Ӏ upregulation, but not with PDE3 or PDE4.

## 4. Discussion

In this study, we investigated the role of AKAPs in TGF-β1/CS-induced EMT in normal human bronchial epithelial BEAS-2B cells and primary HAE cells. We show that the physical interaction between AKAP and PKA is required for TGF-β1-induced EMT, a process that is characterized by reduced E-cadherin and increased collagen I expression. Relevant to the pathophysiology of COPD, CSE similarly induced EMT by stimulation of the release of TGF-β1 and the subsequent activation of TGF-β1 signaling, which was needed for inducing EMT. We found that gene and protein expression of Ezrin, AKAP95, and Yotiao were specifically altered by TGF-β1. Indeed, the single knockdown of Ezrin, AKAP95, or Yotiao diminished TGF-β1-induced collagen Ӏ upregulation, which was further suppressed when they were simultaneously knocked down. Functionally, we report that co-silencing of Ezrin, AKAP95, and Yotiao inhibited TGF-β1-induced cell migration. In addition, the co-silencing of Ezrin, AKAP95, and Yotiao further accelerated the effect of the β_2_-AR, but not of PDE3 or PDE4 on TGF-β1-induced collagen Ӏ upregulation.

Cigarette smoke, as one of the most important inducing factors in COPD, activates the EMT process, which contributes to COPD progression. Sohal et al. demonstrated that reticular basement membrane fragmentation, a key indicator of EMT in vivo, was significantly increased in current smokers with or without COPD as compared to never-smoking control subjects, which also positively correlated with smoking history [7]. Further investigation while using immunohistochemistry indicated that fibroblast specific protein S100A4 was significantly increased in reticular basement membrane clefts in smokers, highlighting the active EMT process in the fragmented reticular basement membrane of smokers and COPD patients [7]. This finding was further confirmed by another in vivo study, in which the significant upregulation of mesenchymal marker vimentin was observed within the small airway epithelium of smokers and COPD subjects [25]. In addition, Milara et al. showed that EMT was increased in primary bronchial epithelial cells of the small bronchi of smokers and COPD patients as compared to the small bronchi of non-smoking control subjects, which indicated that EMT was induced by CS exposure [6]. However, the mechanism of CS-induced EMT in airway epithelial cells is still poorly understood. We show that CS exposure increases TGF-β1 release, which is consistent with a previous study [6]. Additionally, we demonstrate that CS exposure enhanced phosphorylation of SMAD2 and SMAD3 (not shown), which are the key downstream effectors in TGF-β signaling. Furthermore, CS extract-induced E-cadherin loss was inhibited by the pre-treatment with a TGF-β neutralizing antibody. CS exposure also activated cell migration in BEAS-2B cells, which could be decreased when Ezrin, AKAP95, and Yotiao were co-silenced. This indicates that Ezrin, AKAP95, and Yotiao are involved in CS-promoted EMT.

It has been demonstrated that the membrane-cytoskeleton linker Ezrin plays a vital role in cell migration and invasion by modulating the assembly of cytoskeleton elements through Rho GTPase signaling to regulate cytoskeletal organization and cellular phenotypical alterations [26,27,28]. Recent studies have shown that the regulation of Ezrin might be altered in pulmonary diseases. The protein expression of Ezrin was unaltered in the bronchoalveolar lavage fluid of COPD patients [29], while Ezrin protein expression was higher in the epithelium samples from COPD patients as compared to the samples from healthy individuals [30]. Additionally, Ezrin protein was decreased in asthmatic exhaled breath condensate and serum when compared to the non-asthmatic control subjects [31]. Importantly, it was pointed out by several studies that the subcellular location of Ezrin, rather than its expression levels, correlates with its function [32,33]. Such aspects were not studied in BEAS-2B cells yet. Ezrin overexpression was associated with enhanced tumor aggressiveness, while knockdown of Ezrin expression reduced the proliferation, migration, and invasion of cancer cells [34,35,36]. The suppression of Ezrin expression by siRNA prevented the morphological changes, actin filament remodeling, and E-cadherin loss induced by TGF-β1 in alveolar epithelial A549 cells [17]. Surprisingly, in our study, the knockdown of Ezrin decreased TGF-β1-induced collagen Ӏ upregulation, while it did not prevent TGF-β1-induced E-cadherin decrease in BEAS-2B cells, which contrasts to the earlier-mentioned studies using A549 cells [17]. It is important to note that A549 cells are human alveolar basal epithelial cells, while BEAS-2B cells are normal human epithelial cells localizing in bronchus. The specific location in the tissue might partly explain the different observation regarding the function of Ezrin in TGF-β1-induced E-cadherin decrease between our study and those reported earlier.

AKAP95, which is also known as AKAP8, resides in the nucleus and it is involved in DNA replication and controls expression of several proteins that regulate the cell cycle [15,37]. TGF-β1 has been found to increase cell proliferation in the lung structural cells [38,39]. In the current study, we observed that the knockdown of AKAP95 decreased TGF-β1-induced collagen Ӏ production. It is tempting to speculate that this process might be linked to the fact that silencing of AKAP95 inhibited TGF-β1-induced cell proliferation in BEAS-2B cells.

Yotiao, which is also known as AKAP9, is involved in the development and metastasis of several cancers, including breast cancer [40], lung cancer [41], melanomas [42], thyroid carcinomas [43,44], and colorectal cancer [18]. The knockdown of Yotiao by short hairpin RNA inhibited tumor growth in mice, which was partly due to the fact that Yotiao knockdown induced an increase in the epithelial marker E-cadherin and decreased mesenchymal markers N-cadherin and vimentin, indicating that Yotiao plays a crucial role in EMT [18]. We found that siRNA-mediated knockdown of Yotiao significantly decreased TGF-β1-induced upregulation of the mesenchymal marker collagen Ӏ, which indicated that Yotiao acts as a potential target to prevent EMT. Surprisingly, TGF-β1-induced suppression of E-cadherin was further decreased when Yotiao was silenced, which seemed to be conflicting with previous observations [18]. However, earlier we showed that Yotiao and E-cadherin co-localize at the cell membrane of human bronchial epithelial 16HBE cells, highlighting the importance of Yotiao on epithelial barrier function [19]. Together with this finding, our observation in BEAS-2B cells emphasized the importance of Yotiao on cell membrane maintenance.

In the current study, distinct cAMP compartments were differentially activated by the β_2_-agonist fenoterol, the PDE4 inhibitor rolipram, the PDE3 inhibitor cilostamide, and dbcAMP. We found that TGF-β1-induced collagen Ӏ upregulation was suppressed when cAMP donors were applied. Moreover, the co-silencing of Ezrin-AKAP95-Yotiao further decreased TGF-β1-induced collagen Ӏ upregulation by fenoterol, implicating that these AKAPs may diminish the suppression of TGF-β1 induced collagen Ӏ by the β_2_-AR and may, thus, exhibit an inhibitory restraint on its function. β_2_-AR binds to the ezrin/radixin/moesin-binding phosphoprotein 50 (also referred to as NHERF) complex through their PDZ motifs in airway epithelial cells, highlighting the association between Ezrin and β_2_-AR, which is in line with our findings [45]. Intriguingly, the co-silencing of Ezrin-AKAP95-Yotiao silencing did not alter the effects of the PDE4 inhibitor rolipram and the PDE3 inhibitor cilostamide on collagen Ӏ protein expression, which strongly suggested that distinctly AKAP-defined compartments determine the functional outcome of the β_2_-AR when compared to PDE3 and PDE4 in the TGF-β1-induced EMT process (as shown in Figure 13). More investigations are needed to study the difference between β_2_-AR, PDE3, and PDE4. In addition, chronic lung diseases, such as COPD, are discussed in the context of a spectrum of EMT states [46], which would be reflected in our current study by the selective role of Ezrin-AKAP95-Yotiao in diminishing TGF-β1-induced collagen I deposition, without the restoration of E-cadherin, which was reflected by the induction of the EMT-driving transcription factors Snail and Slug.

## 5. Conclusions

Our study demonstrates that the AKAP family members Ezrin, AKAP95, and Yotiao play essential roles in TGF-β1-induced EMT. The co-silencing of Ezrin, AKAP95, and Yotiao inhibits TGF-β1-induced collagen Ӏ production in human bronchial epithelial BEAS-2B cells. Functionally, we show that Ezrin, AKAP95, and Yotiao are required for TGF-β1-induced cell migration, a process that is mimicked by CS. Importantly, Ezrin, AKAP95, and Yotiao seem to act in concert with the β_2_-AR, but not PDE3 or PDE4 in decreasing TGF-β1-induced collagen Ӏ upregulation. CS, as an important risk factor for COPD, induces TGF-β1 release, thereby activating the TGF-β1 signaling pathway, which contributes to EMT progression. Thus, Ezrin, AKAP95, and Yotiao may represent future therapeutic targets for inhibiting bronchial EMT in COPD, possibly in combination with already existing therapies, such as β_2_-agonists and PDE4 inhibitors.

## Figures and Tables

**Figure 1 cells-09-00356-f001:**
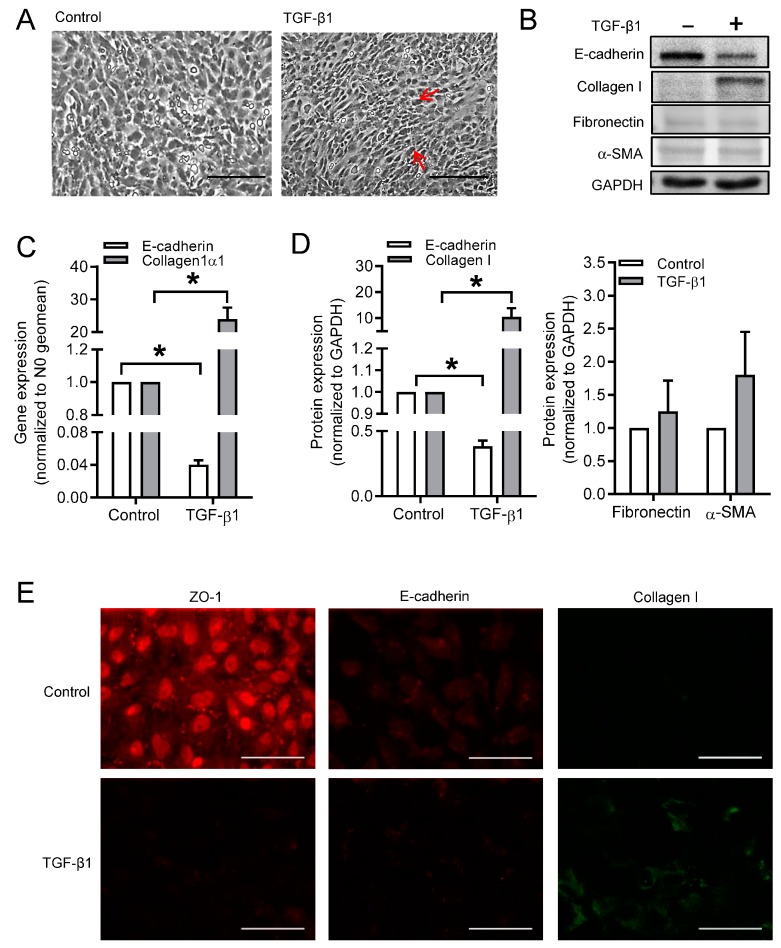
Effects of transforming growth factor-β1 (TGF-β1) on cell morphology and epithelial-to-mesenchymal transition (EMT) markers. (**A**) Morphological changes of BEAS-2B cells after stimulated with TGF-β1 (3 ng/mL) for 24 h. Spindle-like cells were indicated with red arrow. (**B**–**D**) Gene (**B**) and protein (**C**,**D**) expressions of E-cadherin, collagen Ӏ, fibronectin and α-smooth muscle actin (α-SMA) were analyzed in BEAS-2B cells with or without TGF-β1 stimulation using real-time quantitative PCR and western blotting, respectively. Representative western blotting images of E-cadherin, collagen Ӏ, fibronectin and α-SMA were shown in (**C**). (**E**) Immunofluorescence images of ZO-1, E-cadherin, and collagen Ӏ after 24 h stimulation of TGF-β1. Scale bar represents 100 µm. Data represent 3–6 independent experiments. Data are expressed as mean ± SEM, * *p* < 0.05; significant difference between indicated groups.

**Figure 2 cells-09-00356-f002:**
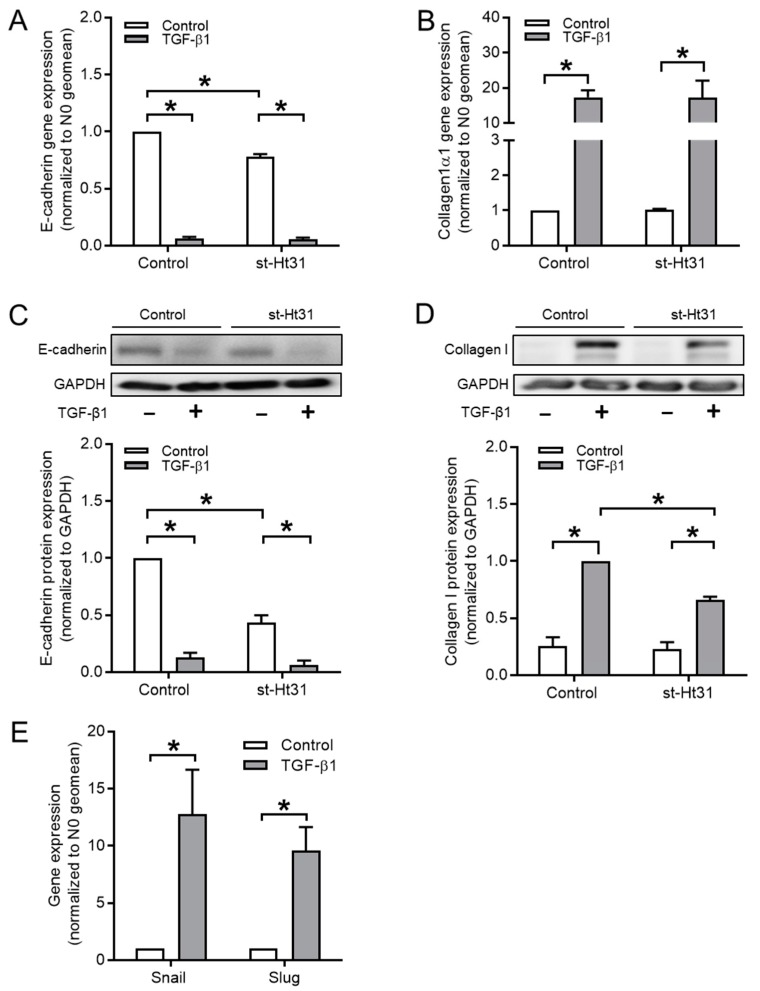
Role of st-Ht31 on EMT markers in BEAS-2B cells. BEAS-2B cells were pre-incubated with 50 µM st-Ht31 for 30 min., following stimulated with 3 ng/mL TGF-β1 for 24 h. Gene (**A**,**C**,**E**) and protein (**B**,**D**) expressions of E-cadherin (**A**,**B**), collagen Ӏ (**C**,**D**), and Snail and Slug were examined while using real-time quantitative PCR and western blotting, respectively. Data represent three independent experiments. Data are expressed as mean ± SEM, * *p* < 0.05; significant difference between indicated groups.

**Figure 3 cells-09-00356-f003:**
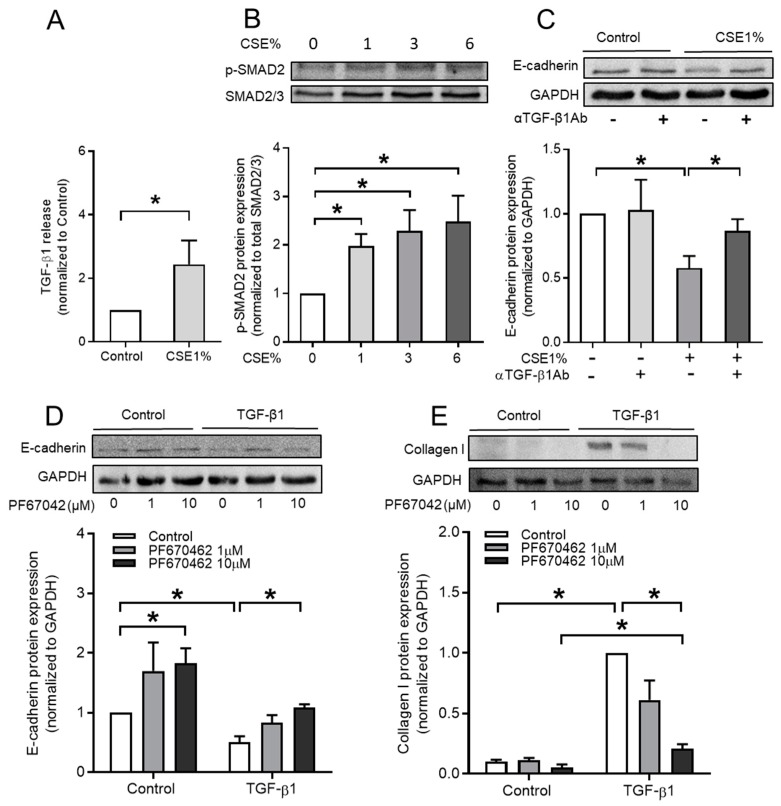
The effect of CS extract on TGF-β1 release, TGF-β1 signaling, and TGF-β1 induced EMT in BEAS-2B cells. The effect of the δ/ε casein kinase I isoform inhibitor PF-670462. (**A**) TGF-β1 release was measured by ELISA using the supernatant after 24 h 1% CSE exposure. (**B**) Representative western blotting images and quantification of phospho-SMAD2 in BEAS-2B cells treated with 24 h CS extract-incubated medium for 1 h. (**C**) BEAS-2B cells were pre-incubated with 10 ng/mL TGF-β neutralizing antibody for 30 min., following stimulated with 1% CSE for 24 h. Representative western blotting images and quantification of E-cadherin were shown. BEAS-2B cells were pre-incubated with 1 µM or 10 µM PF670462 for 30 min., following stimulated with 3ng/mL TGF-β1 for 24 h. Protein (**D**–**E**) expressions of E-cadherin (**D**) and collagen Ӏ (**E**) were examined by western blotting. Data represent 3–6 independent experiments. Data are expressed as mean ± SEM, * *p* < 0.05; significant difference between indicated groups.

**Figure 4 cells-09-00356-f004:**
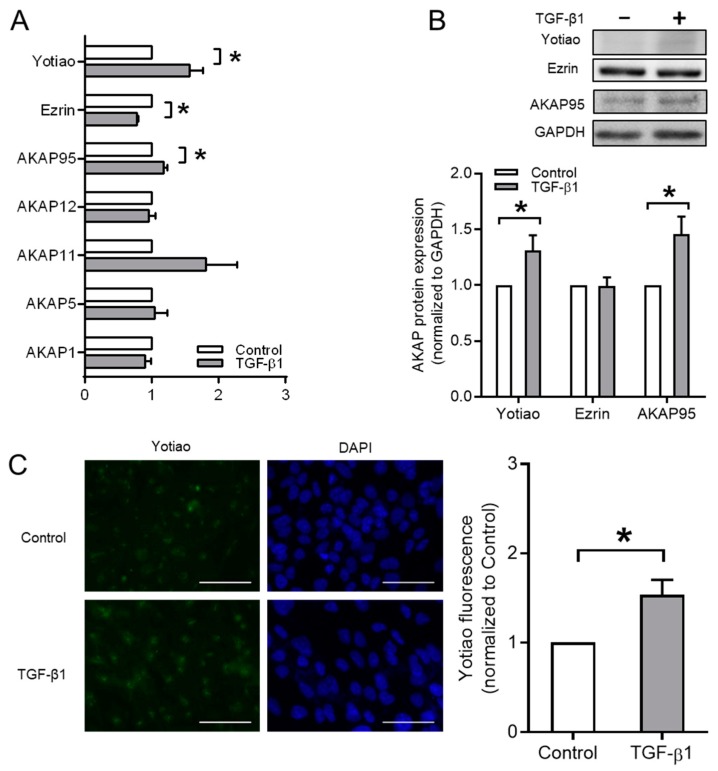
Effect of TGF-β1 on gene and protein expression of A-kinase anchoring proteins (AKAPs) in BEAS-2B cells. (**A**) Gene expressions of *AKAP1*, *AKAP5*, *AKAP11*, *AKAP12*, *AKAP95*, *Ezrin*, and *Yotiao* were studied by real-time quantitative PCR. (**B**) Representative western blotting images and quantification of Yotiao, Ezrin and AKAP95. (**C**) Immunofluorescence images of Yotiao after 3 ng/mL TGF-β1 stimulation for 24 h. Scale bar represents 100 µm. Data represent 5–7 independent experiments. Data are expressed as mean ± SEM, * *p* < 0.05; significant difference between indicated groups.

**Figure 5 cells-09-00356-f005:**
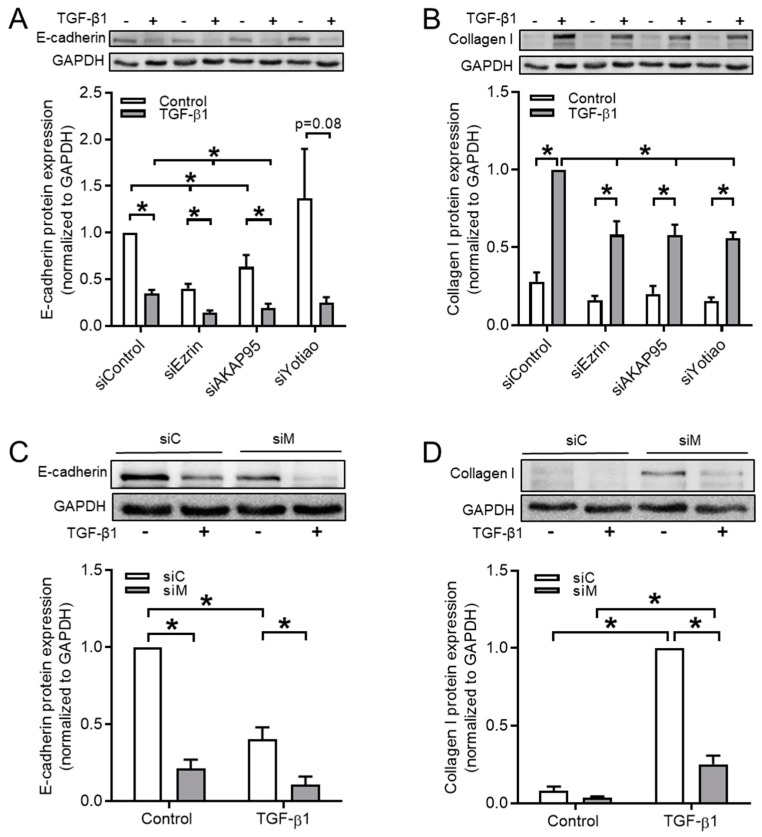
Silencing of Ezrin, AKAP95, and Yotiao diminished TGF-β1-induced collagen Ӏ upregulation. (**A**,**B**) Representative western blotting images and quantification of E-cadherin (**A**) and collagen Ӏ (**B**) in cells transfected with the siRNA of Ezrin, AKAP95 or Yotiao in combination with TGF-β1 treatment. (**C**,**D**) Representative western blotting images and quantification of E-cadherin (**C**) and collagen Ӏ (**D**) in cells transfected with a combination siRNA of Ezrin, AKAP95, and Yotiao together with TGF-β1 treatment. siC, control siRNA; siM, multiple silencing siRNA (Ezrin-AKAP95-Yotiao). Data represent 4–6 independent experiments. The data are expressed as mean ± SEM, * *p* < 0.05; significant difference between indicated groups.

**Figure 6 cells-09-00356-f006:**
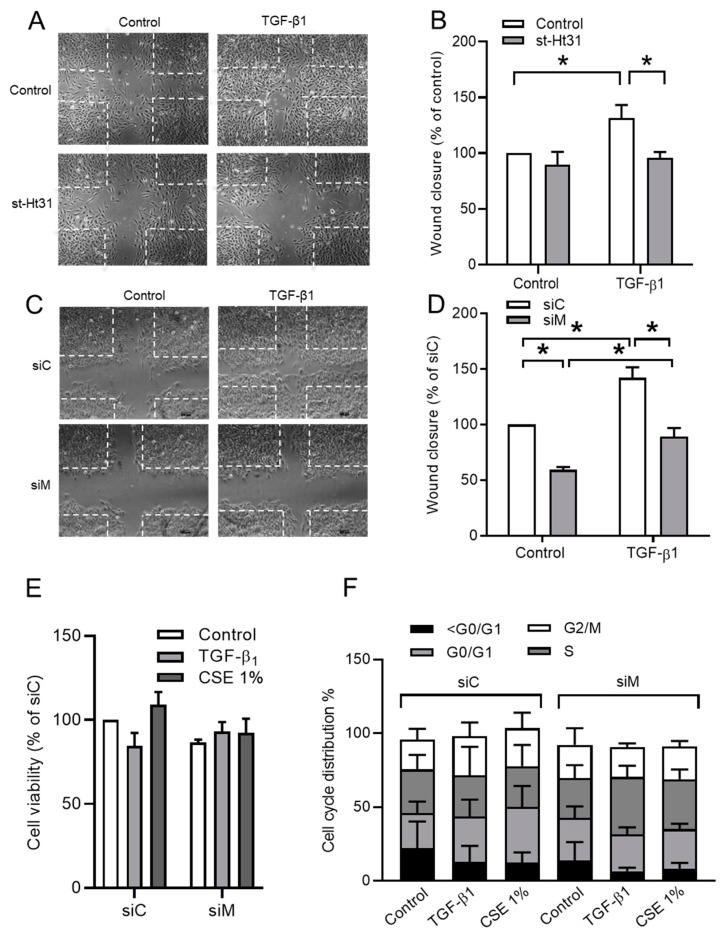
Role of AKAPs—Ezrin, AKAP95, and Yotiao—in TGF-β1-induced cell migration using BEAS-2B cells. (**A**) Representative images of wound healing assay of BEAS-2B cells after 24 h post scratch. The white dotted line indicated borders of scratches at 0 h. (**B**) Quantification of wound closure of cells pre-incubated with 50 µM st-Ht31 for 30 min., following stimulated with 3 ng/mL TGF-β1 for 24 h. (**C**) Representative images of wound healing assay of BEAS-2B cells after 24 h post scratch. The white dotted line indicated borders of scratches at 0 h. (**D**) Quantification of wound closure of TGF-β1 treated cells icombined knockdown of Ezrin, AKAP95, and Yotiao. (**E**) 3-(4,5-dimethylthiazol-2-yl)-2,5-diphenyltetrazolium bromide (MTT) assay in BEAS-2B cells that were transfected with a combination siRNA of Ezrin, AKAP95, and Yotiao together with 3 ng/mL TGF-β1 and 1% CSE exposure. (**F**) Cell cycle distribution in BEAS-2B cells that were transfected with a combination siRNA of Ezrin, AKAP95, and Yotiao together with 3 ng/mL TGF-β1 and 1% CSE exposure. The data represent 3–5 independent experiments. Data are expressed as mean ± SEM, * *p* < 0.05; significant difference between indicated groups.

**Figure 7 cells-09-00356-f007:**
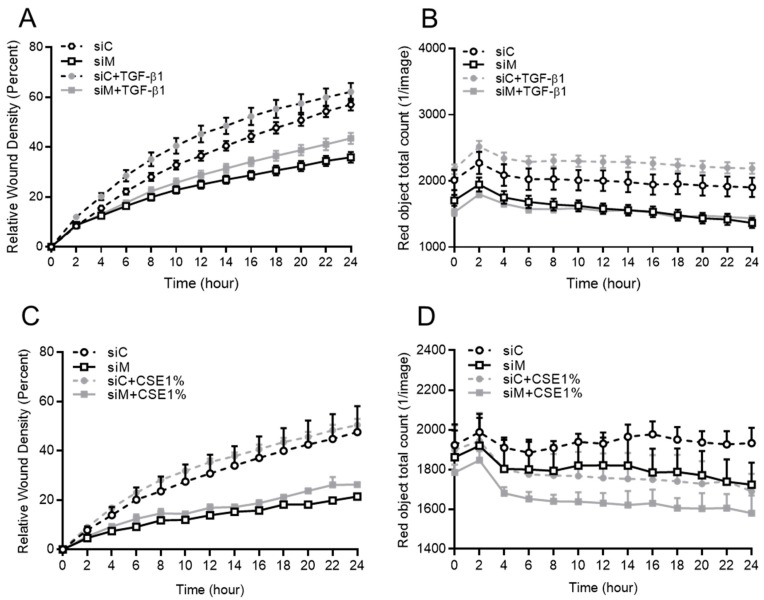
Role of Ezrin, AKAP95, and Yotiao in TGF-β1-induced cell migration while using BEAS-2B cells. (**A**,**C**) Cell migration was monitored every two hours in a 96-well plate using a real time system Incucyte. (**B**,**D**) Cell proliferation was monitored by Incucyte. The BEAS-2B cells were transfected with a combination siRNA of Ezrin, AKAP95, and Yotiao together with 3 ng/mL TGF-β1 and 1% CSE exposure. Data representative of 3–5 independent experiments. The data are expressed as mean ± SEM.

**Figure 8 cells-09-00356-f008:**
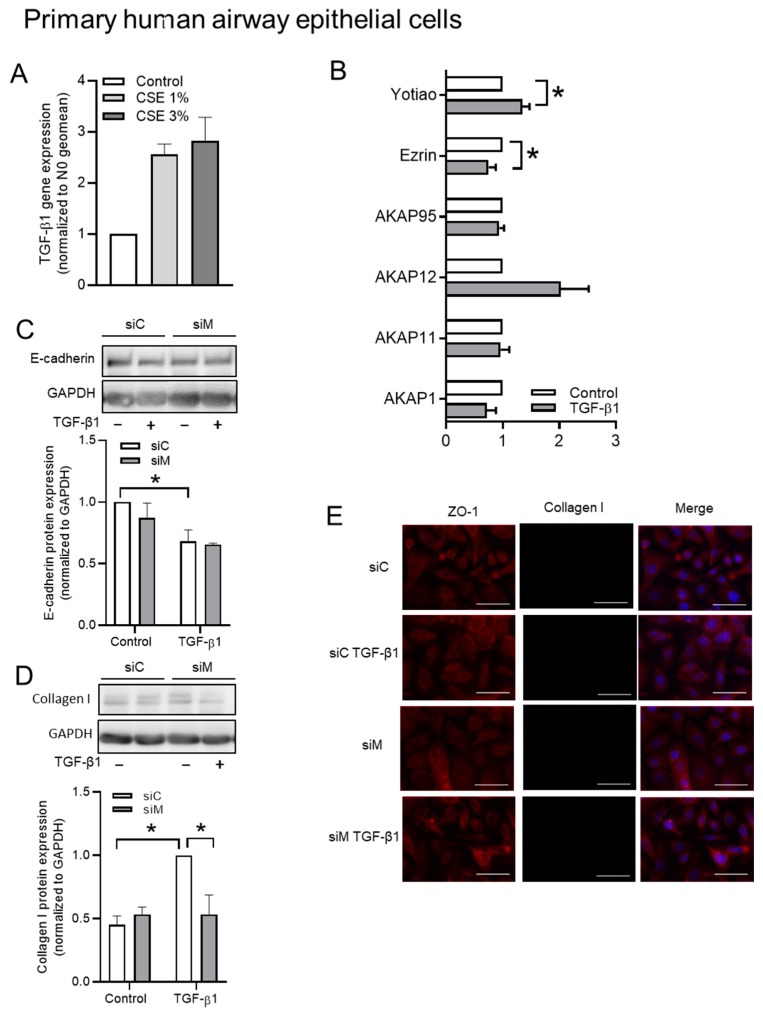
The effect of CS extract on TGF-β1 mRNA release, the effect of TGF-β1 on TGF-β1-induced EMT in pHAE cells. (**A**) TGF-β1 mRNA was measured of cells exposed for 24 h to the indicated CSE concentrations. (**B**) Gene expressions of *AKAP1*, *AKAP11*, *AKAP12*, *AKAP95*, *Ezrin*, and *Yotiao* were studied by real-time quantitative PCR. (**C**,**D**) Protein expressions of E-cadherin and collagen I were analyzed in pHAE cells were transfected with a combination siRNA of Ezrin, AKAP95, and Yotiao together with 3 ng/mL TGF-β1. (**E**) Representative immunofluorescence images of EMT markers in pHAE cells after TGF-β1 stimulation for 24 h. The data represent 3–5 independent experiments. Data are expressed as mean ± SEM, * *p* < 0.05; significant difference between indicated groups.

**Figure 9 cells-09-00356-f009:**
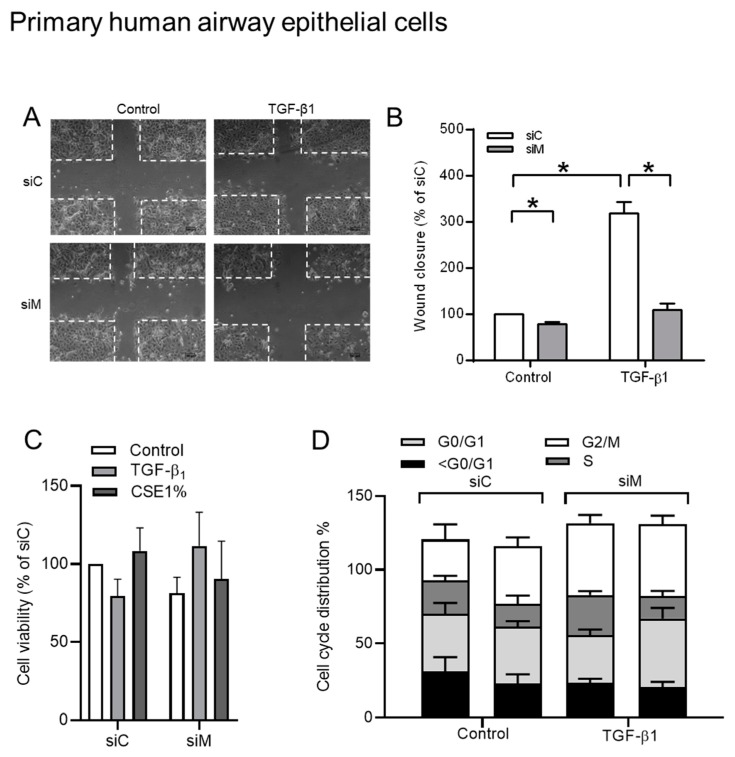
The role of Ezrin, AKAP95, and Yotiao in TGF-β1-induced cell migration while using pHAE cells. (**A**) Representative images of wound healing assay after 24 h post scratch. The white dotted line indicated borders of scratches at 0 h. (**B**) Quantification of wound closure of TGF-β1 treated cells in co-cultured with combined knockdown of Ezrin, AKAP95 and Yotiao. (**C**) MTT assay in pHAE cells that were transfected with a combination siRNA of Ezrin, AKAP95, and Yotiao together with 3 ng/mL TGF-β1. (**D**) Cell cycle distribution in pHAE cells transfected with a combination siRNA of Ezrin, AKAP95 and Yotiao together with 3 ng/mL TGF-β1. Data represent 3–5 independent experiments. The data are expressed as mean ± SEM, * *p* < 0.05; significant difference between indicated groups.

**Figure 10 cells-09-00356-f010:**
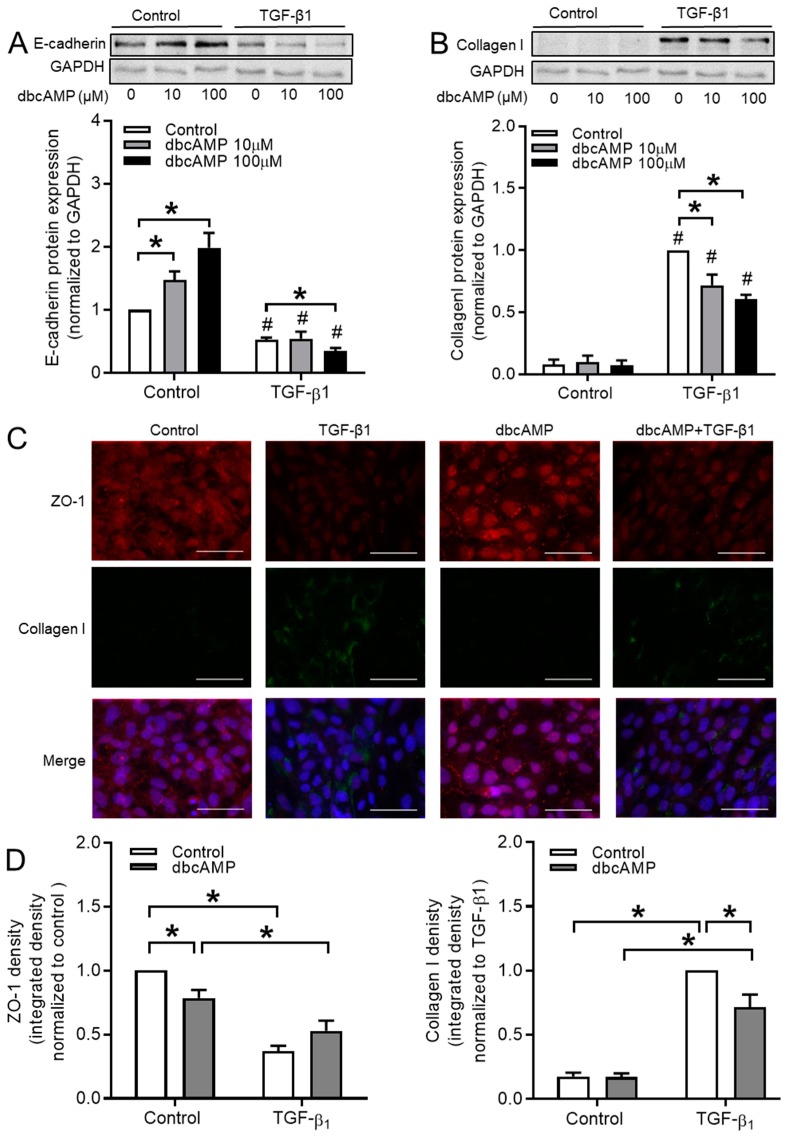
Effect of dbcAMP on TGF-β1-induced EMT markers expression in BEAS-2B cells. (**A**,**B**) protein expressions of E-cadherin (**A**) and collagen Ӏ (**B**) were analyzed in BEAS-2B cells pre-incubated with 10 μM or 100 μM dbcAMP for 30 min. before TGF-β1 stimulation. (**C**,**D**) Immunofluorescence images (**C**) and quantification (**D**) of ZO-1 and collagen Ӏ after TGF-β1 stimulation for 24 h. Scale bar represents 100µm. Data represent 4-6 independent experiments. Data are expressed as mean ± SEM, * *p* < 0.05; significant difference between indicated groups; # *p* < 0.05; significant difference between with or without TGF-β1 treatment.

**Figure 11 cells-09-00356-f011:**
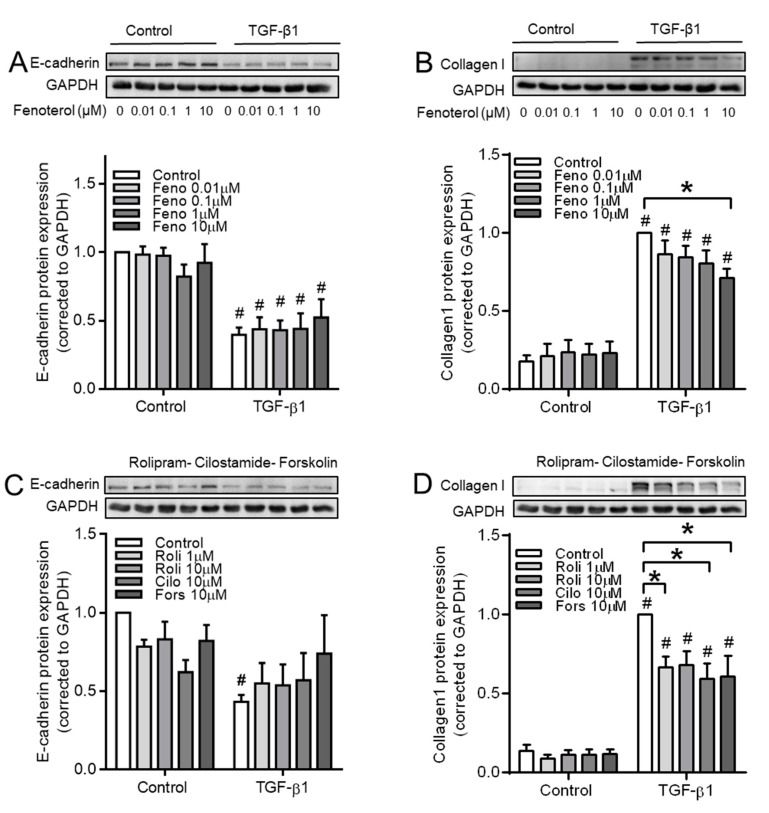
Effect of fenoterol, rolipram, cilostamide and forskolin on TGF-β1-induced EMT makers using BEAS-2B cells. (**A**–**D**) Representative western blotting images and quantification of E-cadherin (**A**,**C**) and collagen Ӏ (**B**,**D**) in cells pre-incubated with fenoterol (**A**,**B**) or rolipram, cilostamide and forskolin (**C**,**D**) before TGF-β1 treatment. Data represent 5–6 independent experiments. Data are expressed as mean ± SEM, * *p* < 0.05; significant difference between indicated groups. # *p* < 0.05; significant difference between with or without TGF-β1 treatment.

**Figure 12 cells-09-00356-f012:**
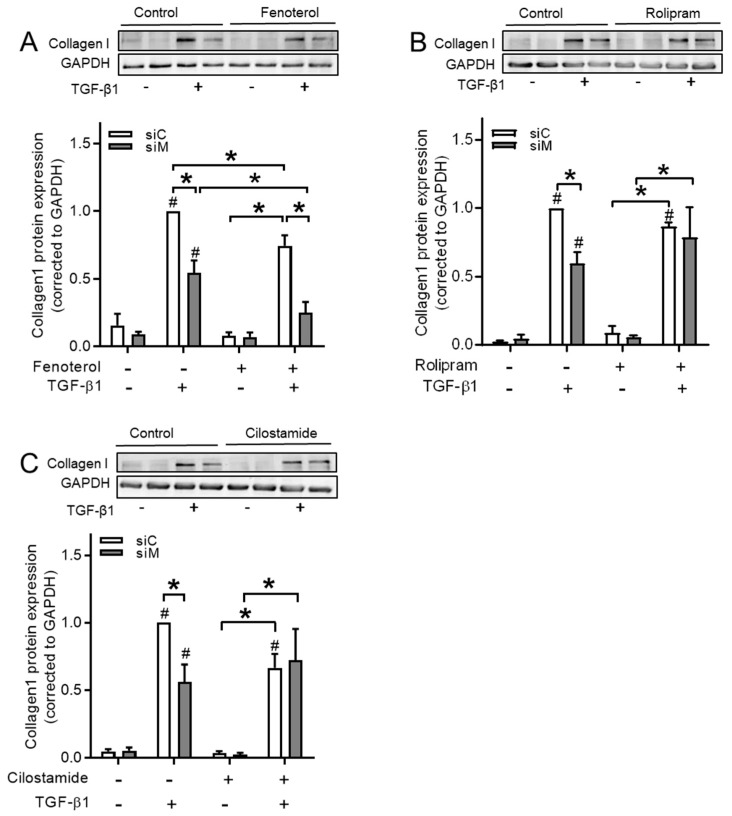
Combined Ezrin, AKAP95 and Yotiao silencing differentially blunted reduction of TGF-β1-induced collagen Ӏ elevation by fenoterol, rolipram and cilostamide in BEAS-2B cells. (**A**–**C**) Representative western blotting images and quantification of collagen Ӏ in cells pre-incubated with fenoterol (**A**) rolipram (**B**) or cilostamide (**C**) before TGF-β1 treatment. Data represent four independent experiments. The data are expressed as mean ± SEM, * *p* < 0.05; significantly different between indicated groups. # *p* < 0.05; significant difference compared to control groups.

**Figure 13 cells-09-00356-f013:**
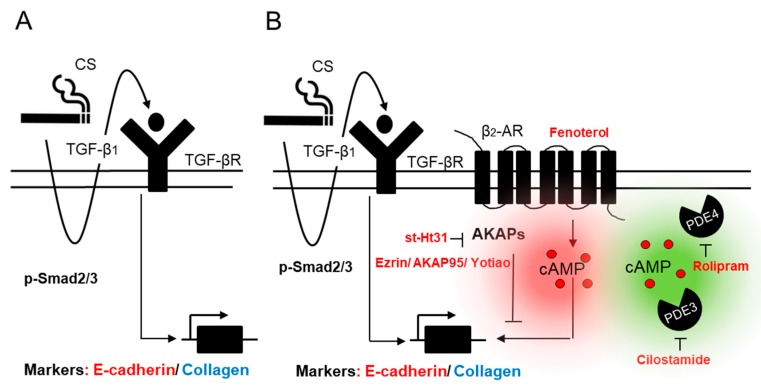
Proposed molecular mechanism explaining the role of different cAMP compartments in TGF-β1/CS-induced EMT in human bronchial epithelial cells. CS exposure induces TGF-β1 release, which in turn activates EMT via SMAD-dependent pathway (**A**). AKAPs, especially focusing on TGF-β1 sensitive AKAPs, Ezrin, AKAP95, and Yotiao, are involved in TGF-β1-induced collagen Ӏ upregulation. Elevation of cAMP by fenoterol, rolipram, and cilostamide differentially diminish TGF-β1-induced *EMT.* Ezrin, AKAP95, and Yotiao seem to be associated with the β_2_-AR but not PDE3 or PDE4 to regulate the suppression of TGF-β1-induced collagen Ӏ upregulation (**B**).

**Table 1 cells-09-00356-t001:** Antibodies used in western blotting and immunofluorescence.

Antibody	Dilution	Company
Anti-E-cadherin	Western blotting, 1:1000 Immunofluorescence, 1:100	BD Biosciences
Anti-ZO-1	Immunofluorescence, 1:100	Invitrogen
Anti-type I collagen-UNLB	Western blotting, 1:1000 Immunofluorescence, 1:20;	SouthernBiotech
Anti-α-SMA	Western blotting, 1:1000	Sigma
Anti-Fibronectin	Western blotting, 1:1000	Santa Cruz Biotechnology
Anti-Ezrin	Western blotting, 1:500	Abcam
Anti-AKAP95	Western blotting, 1:500	Santa Cruz Biotechnology
Anti-Yotiao (Anti-AKAP9)	Immunofluorescence, 1:50 Western Blotting,1:1000	BD Biosciences Novus Biologicals
Anti-p-Smad2	Western blotting, 1:1000	Cell Signaling Technology
Anti-p-Smad3	Western blotting, 1:1000	Cell Signaling Technology
Anti-total Smad2/3	Western blotting, 1:3000	Santa Cruz Biotechnology
Anti-GAPDH Anti-p-PKA Anti-α-Tubulin	Western blotting, 1:10,000 Western Blotting, 1:1000 Western Blotting, 1:3000	Sigma Cell Signaling Technology Sigma

**Table 2 cells-09-00356-t002:** Primer sequences.

Primers	Species	Forward Sequence 5′ to 3′	Reverse Sequence 5′ to 3′
18s	Homo sap.	CGCCGCTAGAGGTGAAATTC	TTGGCAAATGCTTTCGCTC
SDHA	Homo sap.	GGGAAGACTACAAGGTGCGG	CTCCAGTGCTCCTCAAAGGG
RPL13A	Homo sap.	ACCGCCCTACGACAAGAAAA	GCTGTCACTGCCTGGTACTT
E-cadherin	Homo sap.	TGCCCAGAAAATGAAAAAGG	GTGTATGTGGCAATGCGTTC
Collagen 1α1	Homo sap.	AGCCAGCAGATCGAGAACAT	TCTTGTCCTTGGGGTTCTTG
AKAP1	Homo sap.	CCAGTGCAGGAGGAAGAGTATG	CTCCCTCGACACCTCTATCCT
AKAP5	Homo sap.	GACGCCCTACGTTGATCT	GAAATGCCCAGTTTCTCTATG
AKAP11	Homo sap.	CCGGGCTAGTTCTGAATGGG	TGCTCCGACTTCACATCCAC
AKAP12	Homo sap.	CAAGCACAGGAGGAGTTACAG	CTGGTCTTCCAAACAGACAATG
AKAP95	Homo sap.	ATGCACCGACAATTCCGACT	CATAGGACTCGAACGGCTGG
Yotiao	Homo sap.	AACCTGAAGATGTGCCTCCTG	CTGGAGTGCATACCTTTC
Ezrin	Homo sap.	GCTTTTTGATCAGGTGGTAAAGACT	TCCACATAGTGGAGGCCAAAGT
Snail	Homo sap.	ACCACTATGCCGCGCTCTT	GGTCGTAGGGCTGCTGGAA
Slug	Homo sap.	TGTTGCAGTGAGGGCAAGAA	GAGCCTGGTTGCTTCAAGGA
TGF-β1	Homo sap.	GGGACTATCCACCTGCAAGA	CCTCCTTGGCGTAGTAGTCG

**Table 3 cells-09-00356-t003:** The comparison of E-cadherin and Collagen Ӏ protein expression in signal and multiple AKAPs silencing and the pretreatment of st-Ht31.

Treatment	E-cadherin		n	Treatment	Collagen Ӏ		n
	−TGF-β1	+TGF-β1			−TGF-β1	+TGF-β1	
siControl	1.00 ± 0.00	0.35 ± 0.04	6	siControl	0.28 ± 0.06	1.00 ± 0.00	6
siEzrin	0.40 ± 0.05	0.14 ± 0.02	6	siEzrin	0.16 ± 0.03	0.58 ± 0.09	6
siAKAP95	0.63 ± 0.13	0.19 ± 0.05	6	siAKAP95	0.20 ± 0.05	0.58 ± 0.07	6
siYotiao	1.37 ± 0.53	0.25 ± 0.06	4	siYotiao	0.15 ± 0.02	0.56 ± 0.04	4
siControl	1.00 ± 0.00	0.40 ± 0.08	5	siControl	0.08 ± 0.03	1.00 ± 0.00	5
siMultiple	0.21 ± 0.06	0.11 ± 0.05	5	siMultiple	0.04 ± 0.01	0.25 ± 0.06	5
Control	1.00 ± 0.00	0.13 ± 0.02	3	Control	0.26 ± 0.08	1.00 ± 0.00	3
st-Ht31	0.43 ± 0.04	0.06 ± 0.02	3	st-Ht31	0.23 ± 0.06	0.66 ± 0.03	3

**Table 4 cells-09-00356-t004:** The comparison of Collagen Ӏ protein expression in co-silencing multiple AKAPs with or without cyclic adenosine monophosphate (cAMP) donors.

Treatment	Collagen Ӏ					
	Control		siControl		siMultiple	
	−TGF-β1	+TGF-β1	−TGF-β1	+TGF-β1	−TGF-β1	+TGF-β1
Control	0.18 ± 0.04	1.00 ± 0.00				
Fenoterol 10 μM	0.23 ± 0.07	0.71 ± 0.06				
Rolipram 10 μM	0.11 ± 0.03	0.68 ± 0.09				
Cilostamide 10 μM	0.11 ± 0.03	0.59 ± 0.10				
			0.08 ± 0.03	1.00 ± 0.00	0.04 ± 0.01	0.25 ± 0.06
Control			0.16 ± 0.09	1.00 ± 0.00	0.09 ± 0.02	0.54 ± 0.09
Fenoterol 10 μM			0.08 ± 0.02	0.75 ± 0.08	0.07 ± 0.03	0.25 ± 0.08
Control			0.02 ± 0.06	1.00 ± 0.00	0.04 ± 0.02	0.59 ± 0.08
Rolipram 10 μM			0.08 ± 0.05	0.76 ± 0.05	0.05 ± 0.01	0.75 ± 0.22
Control			0.04 ± 0.01	1.00 ± 0.00	0.05 ± 0.02	0.55 ± 0.13
Cilostamide 10 μM			0.03 ± 0.01	0.86 ± 0.29	0.02 ± 0.01	0.72 ± 0.23

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
