# Peer review of "A-Kinase Anchoring Proteins Diminish TGF-β1/Cigarette Smoke-Induced Epithelial-To-Mesenchymal Transition"

_cells, 2020, doi:10.3390/cells9020356_

Round 1
Reviewer 1 Report
This paper reports that AKAPs can control the responses to TGF-beta and cigarette smoke that in airway epithelial cells lead to the pathogenetic mesenchymal transition.
The study is generally convincing; however a few issues need to be addressed.
Major:
Authors generally suggest that cAMP signaling can modulate responses to TGF-beta. However, experiments with TGF-beta together with either knock-down of various AKAPs or cAMP modulators like PDE inhibitors did not include cAMP stimulation. Where is the cAMP supposed to trigger AKAP-mediated responses coming from? Serum? This needs to be fully explained in the results section. In the end of the paper, catecholamine stimulation and beta2 adrenergic receptors (AR) are potentially involved but it is not clear if beta2-AR agonists were used together with TGF-beta (or cigarette smoke). Along the same line, exogenous cAMP is able to reduce TGF-β induced expression of Collagen I, while PDE inhibitors are not. How can this be explained?
Figure 1A An inset with higher magnification would be helpful to appreciate changes in morphology.
Figure 1C: although fibronectin and α-SMA would be relevant markers in characterizing EMT, no difference is detectable after TGF-β treatment (no quantification is provided). This needs to be at least discussed, for example justifying why these markers are no more explored in subsequent experiments.
Figure 2 The authors claim that TGF-β, through Smad pathway, stimulates expression of Collagen I, and that AKAPs are involved in the regulation of this process. However, it is somehow surprising that the effect of Ht31 on Collagen I levels is detectable at the protein level only, but not at the mRNA level. It is indeed known that TGF-β/Smad mediates the induction of TIMPs, that in turn inhibit Collagen-degrading MMPs. Did the authors consider modulation of Collagen I as an indirect effect on MMP expression/activity? Similarly, the contribution of Snail/Slug/Twist should be assessed. These other factors may overall contribute to explain the differential effects detected in E-Cadherin and Collagen I levels in the diverse experimental settings.
Figure 2A Please check label to graph (vertical axis).
Figure 3D and E. Normalization of western blot has been performed on GAPDH but β-actin is shown as a representative image. Amend accordingly.
Paragraph 3.4. How do the authors explain the downregulation of Ezrin after TGF-β ? How this could cope with following experiments using siRNA in which dowregulation of Ezrin has the same (and an additive on top) effect of the inhibition of AKAP95 and Yotiao?
Paragraph 3.5. xCELLigence and Incucyte systems are used to evaluate cell migration. However, these two assays do not seem to provide consistent results. Comparing results obtained with the two techniques is therefore unacceptable, also considering that results from xCELLigence ar not shown. Please provide all data and perform experiments (baseline, TGF-β treatment, siRNAs) with the same system.
Paragraph 3.6. Primary human airway epithelial cells seem to be much more resistant to EMT than immortalized cell lines. Yet, wound closure is significantly increased by TGF-β and reduced after knockdown of siRNAs. Has the contribution of proliferation been excluded also in the context of HAECs?
Discussion The role of CSk1 (Figure 3) should be discussed.
Figure 4C Quantification of immunofluorescence is not acceptable. Please provide Western blot quantification and representative image as for Ezrin and AKAP95.
Minor:
As a general remark, please verify homogeneity of nomenclature when referring to Collagen I throughout the manuscript.
Line 71 “AKAP membranes”
Line 79 Please correct typo and remove «during»
Line 252 “an” should be “and”
Line354 “long” should be “longer”
Line 369 “does” should be “dose”
Figure 1B Please check label to graph (vertical axis).
Figure 5. Clarify the meaning of SiC and SiM in figure legend.
Reviewer 2 Report
Zuo et al., in their manuscript entitled " A-kinase anchoring proteins diminish TGF-β1 / cigarette smoke-induced epithelial-to-mesenchymal transition» report findings suggesting that AKAPs including ezrin, Yotiao and AKAP95 mediate epithelial-to-mesenchymal transition (EMT) induced by TGF-β1 and cigarette smoke in human bronchial epithelial cells. They initially provide evidence that disruption of PKA anchoring decreases E-cadherin but counteracts TGF-β1-induced collagen1 expression. They later show that cigarette smoke increases TGF-β1 release from BEAS-2B cells, enhances signaling through TGF-β receptors, reduces E cadherin expression and promotes cell migration. TGF-β1 alters the expression of ezrin, Yotiao and AKAP95 and silencing of these AKAPs impacts TGF-β1 induced collagen expression and migration of bronchial epithelial cells. Finally, the authors show that co-silencing of ezrin, Yotiao and AKAP95 potentiates the inhibitory effects of the b2-AR agonist fenoterol on TGF-β1-mediated collagen 1 synthesis.
Although investigating the role of AKAPs in cigarette smoke-induced epithelial-to-mesenchymal transition is certainly relevant, my general opinion is that several additional experiments are required to support the conclusions drawn by the authors.
Major issues:
1) In the title the authors state that AKAPs diminish cigarette smoke-induced EMT. This was however not formally demonstrated. To address this issue, the authors should determine the impact of silencing Ezrin, Yotiao and AKAP95 (individually) in primary human airway epithelial cells on collagen 1 protein expression and cell migration induced by cigarette smoke extracts.
2) The functional impact of Ezrin, Yotiao and AKAP95 knockdown is evaluated using a single siRNA. To exclude the possibility of off-target effects at least 2 independent siRNAs should be used for each anchoring protein.
3) Figure 2. The impact of st-Ht31 on cell migration should be determined.
4) The extent of Ezrin, Yotiao and AKAP95 silencing should be evaluated in each experiment. The authors should include Western for Ezrin, Yotiao and AKAP95 in Figure 5,6,7, and 10.
5) Ezrin, Yotiao and AKAP95 regulate multiple cellular functions including, cell cycle progression, DNA duplication, chromatin condensation, Golgi function, cross-linking of plasma membrane protein with the actin cytoskeleton, etc... My concern is that co-silencing of these three AKAPs could potentially affect multiple cellular functions, which could negatively impact cell viability. This aspect should be carefully evaluated because if cells become apoptotic (or pre-apoptotic), they will also stop migrating and producing collagen. This would obviously affect the interpretation of all the experiments presented in Fig 5C and D, Fig. 6, Fig. 7 and Fig 10.
6) Figure 7. TGF-β1 does not appear to modulate the expression of Ezrin, Yotiao and AKAP95 in primary human airway epithelial cells. This is inconsistent with the results presented in Fig.5 (BEAS-2B cell line).
7) Figure 7 should also include collagen 1 and E-cadherin Western blots.
8) Figure 10. Fenoterol reduces TGF-β1-induced collagen 1 production. If ezrin, Yotiao and AKAP 95 were downstream mediators of the effects of b2-adrenergic receptors, then I would have expected that their knockdown would suppress or diminish the inhibitory effect of fenoterol. Surprisingly, the presented data suggest that co-silencing of the three AKAPs amplifies the effect of fenoterol. It is not clear to me how one should interpret this result.
Author Response
From:
Martina Schmidt, PhD
Department of Molecular Pharmacology
University of Groningen
Antonius Deusinglaan 1
9713 AV Groningen, The Netherlands
E-mail: m.schmidt@rug.nl
To:
Sophia Lin
Assistant Editor, MDPI AG
Email: sophia.lin@mdpi.com Groningen January 15, 2020
[Cells] Manuscript ID: cells-641709
Dear Sophie Lin:
Please find enclosed the revision of our manuscript, entitled "A-kinase anchoring proteins diminish TGF-β1/cigarette smoke-induced epithelial-to-mesenchymal transition", by Haoxiao Zuo et al. (Cells-641709).
A detailed description of the changes made in our revised manuscript is given in the point-to-point response below. Importantly, we have performed the majority of the requested additional experiments. Of particular interest, we added several experimental findings in primary human airway epithelial (pHAE) cells. We show that cigarette smoke extract induced the release of TGF-β1 mRNA. We provide evidence that TGF-β1 induced alterations in both E-cadherin and collagen I protein, processes sensitive to the silencing of Ezrin-AKAP95-Yotiao. Likewise, we assessed cell viability and the cell cycle distribution in pHAE cells. Next to pHAE cells, we performed several experiments in BEAS-2B cells including cell viability and cell cyclce distribution. Where possible we added experiments to verify the specificity of the siRNA.
We feel that we have been able to address all remarks adequately and hope that our revised manuscript is now acceptable for publication in the Cells, section Cell Signaling and Regulated Cell Death “New advances in cyclic AMP signaling”. We are looking forward to your response.
With kind regards on behalf of the co-authors,
Martina Schmidt

Reviewer 3 Report
The manuscript by Zuo and colleagues is a very complex paper in which the Authors tried to demonstrate that cigarette smoke may induce alteration in bronchial epithelial cells acting through the involvement of and like TGFβ1. Most experiments are well conducted and correctly explained in the Results section, whereas the comments are not always adequate in the Discussion.
For instance, insufficient explanation is provided for the use of and the different effects caused by st-Ht31 and PF-670462.
At page 19, the Authors report that CS exposure also activates cell migration, which could be decreased by Ezrin, AKAP95 and Yotiao silencing, but this hypothesis is not congruent with data reported in the Results, where they affirm that silencing of those agents tended to decrease CS-induced cell migration without achieving statistical significance. Thus, also the conclusion that Ezrin AKAP95 and Yotiao could be promising therapeutic targets is risky, in my opinion.
Moreover the parallelism between the effects caused by TGFβ1 in BEAS-2B and primary HAE cells is weak. More importantly, is there evidence that the release of TGFβ1 is caused/increased by CS in the latter cells?
Finally, the last figure is not easily understandable, even reading the explanation in the legend, and in my opinion, is not appropriate to summarize the main findings reported in the paper.
Minor points
there are some editing errors (just an example: in the abstract, bronchial epithelial (BEAS-2B, primary HAE cells) probably should be human bronchial epithelial cell lines (BEAS2B) and primary HAE cells. In the fig. 2, panel A, in the label of Y axis there is "NO": what's the meaning?
Author Response

(The authors gave the same response as above.)

Reviewer 4 Report
In this paper, the authors investigated cigarette smoke included TGF-b signaling in EMT and found several potential signaling components involved in. The paper is of interest. I only have a few minor comments.
1. In figure 1A, it is hard to tell the morphological changes based on the figure. I suggest they use a higher magnification to show the morphology of the cells clearly.
2. Most of the western blots they showed in the paper are not labeled properly. They need to label each lane and add more explanation in the figure legends.
Author Response

(The authors gave the same response as above.)

Round 2
Reviewer 1 Report
Authors adequately addressed my concerns
Author Response
From:
Martina Schmidt, PhD
Department of Molecular Pharmacology
University of Groningen
Antonius Deusinglaan 1
9713 AV Groningen, The Netherlands
E-mail: m.schmidt@rug.nl
To:
Billie Jiao
Section Managing Editor
Email: billie.jiao@mdpi.com Groningen January 29, 2020
[Cells] Manuscript ID: cells-641709 Minor Revisions
Dear Billie Jiao:
Please find enclosed the minor revisions of our manuscript, entitled "A-kinase anchoring proteins diminish TGF-β1/cigarette smoke-induced epithelial-to-mesenchymal transition", by Haoxiao Zuo et al. (Cells-641709).
A detailed description of the changes made in our revised manuscript is given in the point-to-point response below. We have adopted the manuscript according to the reviewer suggestions. Importantly, we have changed the Figure 13 to provide a more detailed model about the complex role of A-kinase anchoring proteins in the process of epithelial-to-mesenchymal transition, a process induced by both TGF-β1 and cigarette smoke.
We addressed all remarks and hope that our revised manuscript is now acceptable for publication in the Cells, section Cell Signaling and Regulated Cell Death “New advances in cyclic AMP signaling”. We are looking forward to your response.
With kind regards on behalf of the co-authors,
Martina Schmidt

Reviewer 2 Report
The authors have satisfactorily answered to my main concerns. The manuscript has been significantly improved and, in my opinion, could be accepted for publication.
Author Response

(The authors gave the same response as above.)

Reviewer 3 Report
The paper by Zuo et al. is improved but some perplexities still remain and are the following:
- At page 10, line 291, it is reported that: “Surprisingly, st-Ht31 pretreatment did not prevent …”. In my opinion, given that the same treatment alone significantly decreased E-cadherin gene expression, it is likely that “Consequently, st-Ht31 pretreatment did not prevent..”
- In relation to the Fig. 3, panel A, the Authors report that TGF beta levels in the medium was measured by ELISA, but they forgot to include the method and/or the kit they used to this purpose.
- At page 20 and 21 in relation to the new Fig. 8 they cite “exposure to CSE increases the release of TGF beta1 mRNA” and “The effect of CS extract on TGF-beta1 mRNA release” (a similar sentence also in the text of the legend): did they really measure released mRNA for TGF-beta 1 or did they measure mRNA for TGF-beta1, likely corresponding to modifications in the released TGF-beta1?
- In my opinion, the last figure remains of difficult interpretation. I suggest to divide this figure into two panels: in the former the Authors could draw the effects promoted by TGF-beta1 and CS in the cells; in the latter they could describe the blockade of some of these effects by antagonists (st-Ht31) or cAMP increase.
Minor points:
- Abstract, page 1, line 35: correct the sentence “Epithelial (E-cadherin, ZO-1) and mesenchymal (Collagen I) markers were analyzed as mRNA and proteins”:
- Again in the abstract, there are two antagonists whose role should be defined making the abstract easily understandable. Thus, I would eliminate St-Ht31 disrupted AKAP-PKA interactions at the line 35-36 and I would introduce new words at the lines 40 and 41 in relation to PF-670462 and St-Ht31. I.e. “a process reversed by an inhibitor of ð/epsilon casein kinase I, PF670462” and St-Ht31, AKAP antagonist, decreased…
- At pag 2, line 81, eliminate “during” prior to “is still unclear”
Author Response
The paper by Zuo et al. is improved but some perplexities still remain and are the following:
Major:
Comment C1 (C1): At page 10, line 291, it is reported that: “Surprisingly, st-Ht31 pretreatment did not prevent …”. In my opinion, given that the same treatment alone significantly decreased E-cadherin gene expression, it is likely that “Consequently, st-Ht31 pretreatment did not prevent..”
Response (R1): The text has been changed accordingly.
C2: In relation to the Fig. 3, panel A, the Authors report that TGF beta levels in the medium was measured by ELISA, but they forgot to include the method and/or the kit they used to this purpose.
R2: We apologize for this omission. We added the information into the manuscript (page 4, line 141-142).
C3: At page 20 and 21 in relation to the new Fig. 8 they cite “exposure to CSE increases the release of TGF beta1 mRNA” and “The effect of CS extract on TGF-beta1 mRNA release” (a similar sentence also in the text of the legend): did they really measure released mRNA for TGF-beta 1 or did they measure mRNA for TGF-beta1, likely corresponding to modifications in the released TGF-beta1?
R3: We thank the reviewer for his/her critical remark. We changed the text accordingly.
C4: In my opinion, the last figure remains of difficult interpretation. I suggest to divide this figure into two panels: in the former the Authors could draw the effects promoted by TGF-beta1 and CS in the cells; in the latter they could describe the blockade of some of these effects by antagonists (st-Ht31) or cAMP increase.
R4: We thank the reviewer for his/her critical remark. We changed the figure accordingly.
Minor points:
C1: Abstract, page 1, line 35: correct the sentence “Epithelial (E-cadherin, ZO-1) and mesenchymal (Collagen I) markers were analyzed as mRNA and proteins”:
R1: We thank the reviewer for his/her critical remark. We changed the text accordingly.
C2: Again in the abstract, there are two antagonists whose role should be defined making the abstract easily understandable. Thus, I would eliminate St-Ht31 disrupted AKAP-PKA interactions at the line 35-36 and I would introduce new words at the lines 40 and 41 in relation to PF-670462 and St-Ht31. I.e. “a process reversed by an inhibitor of ð/epsilon casein kinase I, PF670462” and St-Ht31, AKAP antagonist, decreased…
R2: We thank the reviewer for his/her critical remark. We changed the text accordingly.
C3: At page 2, line 81, eliminate “during” prior to “is still unclear”
R3: We thank the reviewer for his/her critical remark. We changed the text accordingly.